Clim. Past Discuss., doi:10.5194/cp-2015-156, 2016 Manuscript under review for journal Clim. Past Published: 15 January 2016 © Author(s) 2016. CC-BY 3.0 License.

This discussion paper is/has been under review for the journal Climate of the Past (CP). Please refer to the corresponding final paper in CP if available.

# Interhemispheric bias in earth's climate response to orbital forcing

# R. Roychowdhury and R. M. DeConto

Department of Geoscience, University of Massachusetts, Amherst, USA

Received: 17 October 2015 - Accepted: 8 December 2015 - Published: 15 January 2016

Correspondence to: R. Roychowdhury (rroychowdhur@geo.umass.edu)

Published by Copernicus Publications on behalf of the European Geosciences Union.

# Abstract

The climate response to orbital forcing shows a distinct hemispheric asymmetry due to the unequal distribution of land in the Northern vs. Southern hemispheres. This asymmetry is examined using a Global Climate Model (GCM) and a Land Hemispheric

- <sup>5</sup> Bias (LHB) is quantified for each hemisphere. The results show how changes in obliquity and precession translate into variations in the calculated LHB. We find that the global climate response to specific past orbits is likely unique and modified by complex climate-ocean-cryosphere interactions that remain poorly known and difficult to model. Nonetheless, these results provide a baseline for interpreting contemporaneous proxy
   <sup>10</sup> climate data spanning a broad range of latitudes, which maybe especially useful in pa-
- leoclimate data-model comparisons, and individual time-continuous records exhibiting orbital cyclicity.

### 1 Introduction

The geographical arrangement of continents on the earth's surface plays a fundamental role in the earth's climate response to forcing. This global "geography" is primarily the result of the horizontal and vertical displacements associated with plate tectonics. While these processes are still in operation, the global continental configuration has been close to its present form since the mid-Cenozoic. Today, more continental land area is found in the Northern Hemisphere (68 %) as compared to the Southern Hemisphere (32 %). These different ratios of land vs. ocean in each hemisphere affect the balance of incoming and outgoing radiation, atmospheric circulation, ocean currents, and the availability of terrain suitable for growing glaciers and ice-sheets. As a result of this land-ocean asymmetry, the climatic responses of the Northern and Southern Hemisphere differ for an identical change in radiative forcing (Barron et al., 1984; Deconto
et al., 2008; Kang et al., 2014).

Multiple studies have shown interhemispheric asymmetry in climate response of Northern and Southern Hemispheres. Climate simulations made with coupled atmosphere-ocean GCMs typically show a strong asymmetric response to greenhouse-gas loading, with Northern Hemisphere high latitudes experiencing more warming compared to Southern Hemisphere high latitudes (Flato and Boer, 2001; Stouffer et al., 1989).

GCMs also show that the Northern and Southern Hemispheres respond differently to changes in orbital forcing (e.g. Philander et al., 1996). While the magnitude of insolation changes through each orbital cycle is identical for both hemisphere, the difference in climatic response can be attributed to the fact that Northern Hemisphere is land-dominated while Southern Hemisphere is water dominated (Croll, 1870). This results

in a stronger response to orbital forcing in the Northern Hemisphere relative to the Southern Hemisphere. In this paper we quantify this bias in the climate response to orbital forcing.

## 15 2 Effect of continental (and oceanic) distribution on climate

10

The changing continental configurations as a result of plate tectonics have been linked with climate change over a wide range of timescales (e.g. Crowley and North, 1996; De-Conto, 2009; Fawcett and Barron, 1998; Hay, 1996). The distribution of continents and oceans has an important effect on the spatial heterogeneity of the Earth's energy balance, primarily via the differences in albedos and thermal properties of land vs. ocean (Trenberth et al., 2009). The latitudinal distribution of land has a dominant effect on zon-ally averaged net radiation balance due to it's influence on planetary albedo and ability to transfer energy to the atmosphere through long-wave radiation, and fluxes of sensible and latent heat. The latitudinal net radiation gradient controls the total poleward heat transport requirement, which is the ultimate driver of winds, and ocean circulation (Stone, 1978).