# Peer review of "Interhemispheric bias in earth's climate response to orbital forcing"

_Climate of the Past, 2015_

## Referee Comment (RC1) · Anonymous Referee #1 · 14 Mar 2016

**1   Summary**

The authors discuss 12 sensitivity experiments with a recent General Circulation Model, called GENESIS3.0. The model is in its atmosphere - slab version (no ocean dynamics). The 12 sensitivity experiments consist in four different idealised orbital configurations (testing different extreme cases of precession and obliquity), repeated three times:

- with modern configuration,
- with an artificial configuration consisting in the modern northern hemisphere mirrored into the southern hemisphere,

[Figure]

- with an artificial configuration consisting in the modern southern hemisphere mirrored into the northern hemisphere.

The purpose of the authors is to document the effect of having, in the other hemisphere, something different than the mirror of the hemisphere being considered. This effect is termed 'Land Hemisphere Bias' (LHB). The authors consider two diagnostic quantities : the "Summer Metric" (essentially a growing-degree-days, or positive-degree-days), and the surface air temperature at 2~m. They conclude that geography has an "important control on the climate system's response to insolation".

**2   Major comments**

There are three major concerns with this study:

- Misleading interpretation. True, the authors correctly point out that what is called the 'LHB' is in fact an effect of the *other* hemisphere being different than a mirror. So, for example, when they say that (p.11) "the Southern Ocean has a perennial positive bias", it is in fact a reference to the fact that the northern hemisphere, with its reactive land masses, increases the temperature of the southern hemisphere compared to a situation where the northern hemisphere would be mainly ocean, like the southern hemisphere... A "bias" in the northern hemisphere is thus a southern-hemisphere effect; but a "bias" in the northern hemisphere is taken by reference to a land-dominated southern hemisphere. Consequently the biases in the southern and the northern hemisphere are calculated with respect to different references. The authors clarify and defend this approach but it may nevertheless be argued that these subtleties obscure the practical implications of these experiments for understanding palaeoclimate records. Consider the following example: "interglacial (warm summer) conditions are muted relative to those of a symmetric

earth". At this stage one should already be aware of the fact that the 'symmetric earth' that we use as a reference differs whether we consider the southern or the northern hemisphere. The authors, yet, go on: "the 'land effect' causes an intensification of 'glacial' vs 'interglacial' conditions in both hemispheres when perihelion coincides with Southern Hemisphere summer". We have the same problem.

The following example might make the case more obvious: "At HIGH obliquity, there exists a negative bias on the Northern Hemisphere Continents". This *bias* is compared to an Earth where southern and northern hemisphere would be dominated by land. Wouldn't we expect a "positive bias" if we were to compare this situation with an earth dominated by oceans?

- Slab ocean. Slab ocean is practical to test different land configurations, but ocean heat transport changes are neglected. This may be acceptable for to studying regional effects (monsoon), but the constrain of fixed heat transport becomes uncomfortable when it comes to study inter-hemispheric effects. At the very least caveats had to be given.

- The focus on temperature response obscures the interest of using an advanced general circulation model for this study. It seems that most if not all the conclusions presented here would have been similarly obtained with an Energy Balance Model. Nothing is said about atmospheric connections, hydrological budget, pressure response. Possible implications for moisture transport are evoked in the conclusions but they are not backed by proper analysis.

**3   Minor comments**

There are a few places with loose wording. E.g.:

- p. 4 line 19 : "should theoretically move equally north and south according to the

hemisphere experiencing summer". The theory precisely says that it should not be symmetric ! (given the asymmetry)

- p. 10 line 15 : "perennial positive bias" : in what sense is it perennial ?
- Use standard definitions when they exist. E.g.: "orbital precession" is in fact the heliocentric longitude of the perihelion.

References to literature:

- Recent landmark citations (Huybers (2006), Raymo (2006)) are valid, but proper credit should be given to Pollard for the use of PDD, and Berger for pioneering analysis of insolation.
- The introduction quotes earlier works on effects associated with Earth's geography on climate sensitivity, but little is said about how the results presented here compare with earlier findings.

Notation clash:

- $S$ is used both for the summer insolation metric, and for the south-symmetric simulation.

**4 Conclusion**

This review has questioned the relevance of the experiments, their implications for our understanding of palaeoclimates dynamics, and the depth of the analysis, without being able to conclude on a positive recommendation for publication.

---

## Referee Comment (RC2) · Anonymous Referee #2 · 27 Mar 2016

This paper studies the climate impact of interhemispheric asymmetry of the land mass in a coupled AGCM-slab ocean model, with the focus on the potential impact on ice sheet melting, by using indexes of accumulated threshold insolation/temperature. In addition to the control run, two experiments are performed with the symmetric land mass using the present NH and SH land mass, respectively. The topic and experimental design are interesting. However, the paper is not well presented, the discussion and conclusion are ambiguous. I will not recommend the paper for publication unless it goes through a major revision.

Major comment: My major criticism is the way the discussion is presented on the S, J, and b̂ indexes. These indexes are not easy to understand, because of the threshold cut off and the time accumulation, and therefore the mechanism for their response patterns are not always straightforward (e.g. Figs.3-6). At least, the authors should

present and discuss the global climate response in terms of the basic variables, such as temperature, before discussing the corresponding threshold index, say, S. As it stands, it is very difficult to follow the discussion and understand the pattern of index response is. For example, in Fig.5, why the sign is negative over land, and why the sign is reversed over the ocean?

Minor comments: 1) I don't like the word "bias" here. Bias, at least to a climatologist, usually implies some systematic error (from some truth). Here, the LHB really refers to the potential climate impact of land mass of each hemisphere, and there is no error involved. It is just some idealization. (save a serious comparison with paleo world). I think "impact" or "effect" or some other words, will be much better than "bias".

2) The authors should highlight one serious caveat in their study, the slab ocean, which assumes a constant ocean heat transport such that the readers should realize the paper is studying an idealized land hemispheric effect (or bias if they call it) in an idealized coupled world. This is important for two reasons. First, the slab ocean works only for short time scales. For paleoclimate application (as the paper is intended for), however, it is the final long term impact that matters. The long term impact can depend critically on the ocean circulation and can differ dramatically from that derived from slab ocean model. Second, due to Bjerkness compensation, the ocean heat transport usually will change in response to climate forcing.

3) Fig.2, caption: seems to be of wrong sign in (b), from low to high obliquity i.e. (b) is high obliquity – low obliquity. Please clarify. Partly, this reflects the lack of discussion mechanism of the pattern of the index as discussed above.

---

## Author Comment (AC1) · 1 May 2016

**Response to Anonymous Referee 1**

May 3, 2016

We thank the two anonymous reviewers for providing thorough and constructive feedback on our manuscript. In the following section, we respond to the specific points raised by each reviewer:

**1 Major Comment #1**

*Misleading interpretation. True, the authors correctly point out that what is called the 'LHB' is in fact an effect of the other hemisphere being different than a mirror. So, for example, when they say that (p.11) "the Southern Ocean has a perennial positive bias", it is in fact a reference to the fact that the northern hemisphere, with its reactive land masses, increases the temperature of the southern hemisphere compared to a situation where the northern hemisphere would be mainly ocean, like the southern hemisphere. . . A "bias" in the northern hemisphere is thus a southern-hemisphere effect; but a "bias" in the northern hemisphere is taken by reference to a land-dominated southern hemisphere. Consequently the biases in the southern and the northern hemisphere are calculated with respect to different references. The authors clarify and defend this approach but it may nevertheless be argued that these subtleties obscure the*

*practical implications of these experiments for understanding palaeoclimate records. Consider the following example: "interglacial (warm summer) conditions are muted relative to those of a symmetric earth". At this stage one should already be aware of the fact that the 'symmetric earth' that we use as a reference differs whether we consider the southern or the northern hemisphere. The authors, yet, go on: "the 'land effect' causes an intensification of 'glacial' vs 'interglacial' conditions in both hemispheres when perihelion coincides with Southern Hemisphere summer". We have the same problem. The following example might make the case more obvious: "At HIGH obliquity, there exists a negative bias on the Northern Hemisphere Continents". This bias is compared to an Earth where southern and northern hemisphere would be dominated by land. Wouldn't we expect a "positive bias" if we were to compare this situation with an earth dominated by oceans?*

**Author Response:** We appreciate the reviewer's careful reading of our manuscript, however, it is not clear to us, what point the reviewer is making. To clarify, the point of the paper is to identify the effect of asymmetry in the hemispheric land distribution of the earth on the climate response of each hemisphere.

When we are calculating the bias for NH, we are trying to identify the effect of southern hemisphere land distribution on Northern Hemisphere climate. We do this by comparing the climate response of NH in a real world, to a hypothetical symmetrical world, in which SH is also land dominated. To study the effect of the opposite hemisphere (in this situation, the opposite hemisphere is SH), we need to keep the hemisphere in question (NH here) constant. By doing the experiments twice, once with real (asymmetric) SH, and once with hypothetical (symmetric) SH, and keeping the NH land distribution constant in both experiments, we isolate the effect of water dominated SH on NH climate response. Similarly, to calculate the bias for SH, we keep the SH land distribution constant in both experiments, and run the simulations twice, once with real (asymmetric) NH, and once with hypothetical (symmetric) NH.

There is no ambiguity in our experiment design, and there exists ONLY ONE possible

symmetrical reference model for each hemisphere. Thus, when we calculate the bias for NH, there can be only one symmetrical reference (North-Symmetric), which can be used to study the effect of SH on NH climate. It is not possible to use the South-Symmetric reference model to calculate the bias in NH, because the first condition, i.e. the hemisphere being considered should be same, is invalidated! Fig 1 and Fig 2 (In the supplement) explains this graphically.

The climate response of the Earth at each latitude is different in the Northern and Southern Hemisphere. This difference in the climate response can be broken down into three components:

1. The insolation intensity depending on precession

2. The effect of NH land configuration on NH climate response (and vice-versa)

3. The effect of SH land configuration on NH climate response (and vice-versa)

In our paper, we examine the third point, i.e. the effect of SH land configuration on NH climate response, and vice-versa. To achieve our objective, it is imperative that we compare the NH 'Land Asymmetry Effect' by comparing the NH climate with a land dominated symmetric earth, and vice versa. Thus, when we say "At HIGH obliquity, there exists a negative bias on the Northern Hemisphere Continents", we are indeed comparing the climate of the Northern Hemisphere with a hypothetical earth in which the Southern Hemisphere is also land dominated, so that we can isolate out the effect which Southern Hemisphere land distribution has on the Northern Hemisphere.

As suggested by Referee 2, we will rephrase the "Land Hemispheric Bias" as "Land Asymmetry Effect". To reiterate, the 'LAE' thus quantifies the effect of the asymmetry in the opposite hemisphere, by systematically comparing the response of the real world, to the response of 1) a symmetric world with mirrored northern hemispheres, and 2) a symmetric world with mirrored southern hemispheres.

*"The following example might make the case more obvious: "At HIGH obliquity, there*

*exists a negative bias on the Northern Hemisphere Continents". This bias is compared to an Earth where southern and northern hemisphere would be dominated by land. Wouldn't we expect a "positive bias" if we were to compare this situation with an earth dominated by oceans?"*

Again, we'll attempt to clarify. To calculate our 'Effect' (earlier termed 'bias') in the Northern Hemisphere, we change the topography of the Southern Hemisphere. Thus, we compare Northern Hemisphere climate of the 'real world' with a symmetrical reference, which has land dominated SH topography (Fig 1).

1. To estimate our 'Land Asymmetry Effect' for the Northern Hemisphere, we must keep the NH land topography constant in our comparisons. Thus, we cannot have an ocean dominated Northern Hemisphere if we are trying to estimate the 'Land Asymmetry Effect' for the Northern Hemisphere.

2. If we compare the NH climate with an earth that has only ocean in the Southern Hemisphere (no land mass, i.e. only ocean), we would get a stronger negative bias, and not a positive bias.

We propose to revise our paper to make our experimental objective clearer to the reader, and replace the usage of the term "Land Hemispheric Bias" with "Land Asymmetry Effect". The practical implications of these results are many, such as the following observation: During the late Pliocene- early Pleistocene, the Northern Hemisphere and Southern Hemisphere integrate summer insolation metrics varied at both obliquity and precessional frequencies. However, the Northern Hemisphere had stronger precessional amplitude than the Southern Hemisphere, which can be ascribed to the "Land Asymmetry Effect" (Manuscript in preparation).

[Figure]

**2   Major Comment #2**

*Slab ocean. Slab Ocean is practical to test different land configurations, but ocean heat transport changes are neglected. This may be acceptable for to studying regional effects (monsoon), but the constrain of fixed heat transport becomes uncomfortable when it comes to study inter-hemispheric effects. At the very least caveats had to be given.*

**Author Response:** We agree with the reviewer here, but we think the reviewer might have missed this in the paper. This issue is addressed explicitly in two places in our paper. First, in the methods section where we state: "We use the latest (2012) version of the Global ENvironmental and Ecological Simulation of Interactive Systems (GENESIS) 3.0 GCM with a slab ocean component (Thompson and Pollard, 1997) rather than a full-depth dynamical ocean (Alder et al., 2011). The slab-ocean version of the GCM allows numerous simulations with idealized global geographies and greatly simplifies interpretations of the sensitivity tests by precluding complications associated with ocean model dependencies."

At the end of the paper, we state: "Future work should include complimentary simulations with AOGCMs, to explore the potential modifying role of ocean dynamics on the LHB, not accounted for here."

Furthermore, the actual ocean heat transport in the slab component is not fixed, but changes relative to the land-ocean fraction in each band of latitude and as a function of the local temperature gradient. While slab-ocean GCMs clearly have limitations, the use of a fully coupled AOGCM would be computationally impractical for this exercise, and would add additional, likely model-dependent, complexities to this first analysis of the LHB issue.

**3 Major Comment #3**

*The focus on temperature response obscures the interest of using an advanced general circulation model for this study. It seems that most if not all the conclusions presented here would have been similarly obtained with an Energy Balance Model. Nothing is said about atmospheric connections, hydrological budget, and pressure response. Possible implications for moisture transport are evoked in the conclusions but they are not backed by proper analysis.*

**Author Response:** : We agree with the referee in principle. However, we use a GCM in this study, because it allows a more realistic spatial analysis (including the influence of atmospheric dynamics) that would be precluded by simple EBMs. We view the fact that the general conclusions would likely be supported by a simple EBM as a good thing, supporting the veracity of the overall conclusions. We also note, that EBMs have been replaced as the standard tool for the modern paleoclimatologists in data-model comparison exercises. One goal of this paper is to provide some guidance on the strength of the LHB, when interpreting paleoclimate data on orbital timescales.

We feel that an in-depth dynamical analysis is beyond the scope of this initial investigation of the issue. However, some simple analysis of orbital controls on meridional moisture transports and precipitation could be added in a revised manuscript.

**4 Minor Comments**

*p. 4 line 19 : "should theoretically move equally north and south according to the C3 CPD Interactive comment Printer-friendly version Discussion paper hemisphere experiencing summer". The theory precisely says that it should not be symmetric! (given the asymmetry)*

**Author Response:** We agree. What we meant was "should ideally move equally north and south according to the hemisphere experiencing summer". We will correct this in future.

*p. 10 line 15 : "perennial positive bias" : in what sense is it perennial ?*

**Author Response:** Perennial positive bias refers to positive bias present at all orbital configurations, i.e. at all obliquity and precessional configurations.

*1)* Use standard definitions when they exist. E.g.: "orbital precession" is in fact the heliocentric longitude of the perihelion.

**Author Response:** We agree that the standard definition should be used instead.

**5   References to Literature**

*Recent landmark citations (Huybers (2006), Raymo (2006)) are valid, but proper credit should be given to Pollard for the use of PDD, and Berger for pioneering analysis of insolation.*

**Author Response:** We will cite Pollard in our revised paper. We have cited Berger in our paper.

*The introduction quotes earlier works on effects associated with Earth's geography on climate sensitivity, but little is said about how the results presented here compare with earlier findings.*

**Author Response:** The climate response of the Earth at each latitude is different in the Northern and Southern Hemisphere. This difference in the climate response can be broken down into three components:

1. The insolation intensity depending on precession 2. The effect of NH land configuration on NH climate response (and vice-versa) 3. The effect of SH land configuration on NH climate response (and vice-versa)

The earlier works, which have been cited in the introduction, refer to the 1st and 2nd points, or the effect of insolation and Earth's local geography on climate sensitivity. In our study, we focus on the 3rd point, i.e. the effect of the geography of the opposite hemisphere on climate sensitivity. The results communicated here are different from any kind of previous study, and hence we have not compared our results with earlier findings.

**Supplement:**

**1) To calculate the bias in Northern Hemisphere:**
   (i.e. effect of SH land distribution on NH climate response)

Northern Hemisphere (Hemisphere being considered) is constant

[Figure]

- ➤ Two experiments are run: (A)Real world (B) North-symmetric world
- ➤ Climate response in the Northern Hemisphere in (A) and (B) are different.
- ➤ This difference in climate response (bias) is caused due to the different land distribution in the Southern Hemisphere in experiments (A) and (B).
- ➤ Thus, **LHB$_{NORTH-HEMISPHERE}$ = climate response of NH in (A) –climate response of NH in (B)**

2) *Now, if we try using South-symmetric world to calculate the bias in NH:*

NH is different! Hence cannot be used to calculate the LHB$_{NORTH-HEMISPHERE}$.

[Figure]

*From the above, it should be clear why we cannot use South-symmetric model to calculate the bias for Northern Hemisphere.*

**2) To calculate the bias in Southern Hemisphere:**

(i.e. effect of NH land distribution on SH climate response)

[Figure]

Southern Hemisphere (Hemisphere being considered) is constant

- ➤ Two experiments are run: (A)Real world (C) South-symmetric world
- ➤ Climate response in the Southern Hemisphere in (A) and (C) are different.
- ➤ This difference in climate response (bias) is caused due to the different land distribution in the Northern Hemisphere in experiments (A) and (C).
- ➤ Thus, **LHB$_{SOUTH-HEMISPHERE}$ = climate response of SH in (A) –climate response of SH in (C)**

2) *Now, if we try using North-symmetric world to calculate the bias in SH:*

SH is different! Hence cannot be used to calculate the LHB$_{SOUTH-HEMISPHERE}$

[Figure]

*From the above, it should be clear why we cannot use North-symmetric model to calculate the bias for Southern Hemisphere.*

---

## Author Comment (AC2) · 1 May 2016

**Response to Anonymous Referee #2**

May 1, 2016

We thank the two anonymous reviewers for providing thorough and constructive feed-back on our manuscript. In the following section, we respond to the specific points raised by each reviewer:

**1   Major Comment #1**

My major criticism is the way the discussion is presented on the S, J, and bËĘ indexes. These indexes are not easy to understand, because of the threshold cut off and the time accumulation, and therefore the mechanism for their response patterns are not always straightforward (e.g. Figs.3-6). At least, the authors should present and discuss the global climate response in terms of the basic variables, such as temperature, before discussing the corresponding threshold index, say, S. As it stands, it is very difficult to follow the discussion and understand the pattern of index response is. For example, in Fig.5, why the sign is negative over land, and why the sign is reversed over the ocean?

**Author Response:** Our primary motivation for working with the Total Integrated Summer Insolation (S) and Summer Energy (J) metrics is that these are more robust indicators of the melting of ice-sheets, as compared to simpler indices like summer temperature or insolation intensity. The S and J indices take into account both the insolation intensity, as well as the duration of the melt season. Hence, a positive change in the S or J index can mean an increase in insolation intensity, or an increase in melt season duration, or both (similarly, a decrease in S can mean a decrease in insolation intensity, or melt duration, or both). However, we agree that these indices may not be easy to understand, because of their non-linear response to insolation changes. To provide additional clarification, we will revise our paper to include a more thorough discussion in terms of the basic variables like summer temperature, which will be followed by our discussion in terms of S and J indexes.

In Fig 5., we plot the estimated 'Land Hemispheric Bias' in surface temperatures for present day continental geography and orbit. The negative sign over land (Northern Hemisphere) means that the Northern Hemisphere land surfaces have a lower surface temperature when compared to a symmetric earth, in which both hemispheres are land dominated. Similarly, the positive signs over water imply that the surface temperatures are higher when compared to a symmetric earth.

**2   Minor Comment #1**

*I don't like the word "bias" here. Bias, at least to a climatologist, usually implies some systematic error (from some truth). Here, the LHB really refers to the potential climate impact of land mass of each hemisphere, and there is no error involved. It is just some idealization. (save a serious comparison with paleo world). I think "impact" or "effect" or some other words, will be much better than "bias".*

**Author Response:** We agree that the word "bias" brings about different connotations here. We propose to remove the word "bias" all together, and replace "Land Hemispheric Bias" with "Land Asymmetry Effect' (LAE).

**3   Minor Comment #2**

*The authors should highlight one serious caveat in their study, the slab ocean, which assumes a constant ocean heat transport such that the readers should realize the paper is studying an idealized land hemispheric effect (or bias if they call it) in an idealized coupled world. This is important for two reasons. First, the slab ocean works only for short time scales. For paleoclimate application (as the paper is intended for), however, it is the final long term impact that matters. The long term impact can depend critically on the ocean circulation and can differ dramatically from that derived from slab ocean model. Second, due to Bjerkness compensation, the ocean heat transport usually will change in response to climate forcing*

**Author Response:** We agree with the reviewer here, this issue is addressed explicitly in two places in our paper. First, in the methods section where we state: "We use the latest (2012) version of the Global ENvironmental and Ecological Simulation of Interactive Systems (GENESIS) 3.0 GCM with a slab ocean component (Thompson and Pollard, 1997) rather than a full-depth dynamical ocean (Alder et al., 2011). The slab-ocean version of the GCM allows numerous simulations with idealized global geographies and greatly simplifies interpretations of the sensitivity tests by precluding complications associated with ocean model dependencies.".

At the end of the paper, we state: "Future work should include complimentary simulations with AOGCMs, to explore the potential modifying role of ocean dynamics on the LHB, not accounted for here."

Furthermore, the actual ocean heat transport in the slab component is not fixed, but changes relative to the land-ocean fraction in each band of latitude and as a function of the local temperature gradient. While slab-ocean GCMs clearly have limitations, the use of a fully coupled AOGCM would be computationally impractical for this exercise, and would add additional, likely model-dependent, complexities to this first analysis of the LHB issue.

**4 Minor Comment #3**

*Fig.2, caption: seems to be of wrong sign in (b), from low to high obliquity i.e. (b) is high obliquity – low obliquity. Please clarify. Partly, this reflects the lack of discussion mechanism of the pattern of the index as discussed above.*

**Author Response:** We agree that our wording in the caption is confusing, and we will correct this in future. For example, for HIGH and LOW obliquity experiments, we run our GCM simulations with the highest and lowest obliquity values. In Fig 2c and 2d, we show the difference in summer energy (J) between the HIGH and LOW obliquity experiments. HIGH Obliquity experiment has a higher Summer Energy (J) in the high latitudes than the LOW Obliquity experiment; hence our HIGH-LOW figure shows positive values in the high latitudes. This is a great point, and we will fix our wordings in the revised manuscript.